# LCST-UCST Transition Property of a Novel Retarding Swelling and Thermosensitive Particle Gel

**DOI:** 10.3390/ma16072761

**Published:** 2023-03-30

**Authors:** Liang Li, Jixiang Guo, Chuanhong Kang

**Affiliations:** Enhanced Oil Recovery Institute, China University of Petroleum, Beijing 102249, China

**Keywords:** water plugging agent, retarding swelling, thermosensitive particle gel, the upper critical solution temperature (UCST), hydrophobically modified poly(vinyl alcohol)

## Abstract

Super absorbent resin particles used as profile control and water plugging agent remains a deficiency that the particles swells with high speed when absorbing water, resulting in low strength and limited depth of migration. To address this issue, we proposed a thermosensitive particle gel possessing the upper critical solution temperature (UCST), which was synthesized from hydrophobically modified poly(vinyl alcohol)s (PVA) with glutaraldehyde (GA) as a cross-linker. The structure of the hydrogel was characterized by Fourier transform infrared spectrophotometer (FTIR) and nuclear magnetic resonance (NMR). The thermosensitive-transparency measurement and swelling experiment show that the hydrophobic-modified PVA solutions and corresponding hydrogels exhibited thermosensitive phase transition behaviors with lower critical solution temperature (LCST) and UCST. The results indicated that the temperature-induced phase transition behavior of CHPVA hydrogels leads to their retarding swelling property and great potential as an efficient water plugging agent with excellent temperature and salt resistance.

## 1. Introduction

Water flooding is the production method of most oil fields, and profile control is one of the important techniques to improve water flooding recovery [1,2,3]. Preformed particle gels (PPG) have played an important role in deep profile control in recent years, and have been widely applied in many oil fields at home and abroad [4,5,6,7]. However, there is a problem of limited injection depth due to the high water swelling speed of conventional PPGs. In addition, the strength of particles after swelling is so weak that they are easy to crush. Although many domestic researchers try to delay the swelling rate by varied methods, little progress is obtained till now [8]. Therefore, there is an emerging interest in particle gels that retard swelling and possess favorable mechanical. There are two broad classes to achieve slow expansion of gel particles. The first one is physical wrapping achieved through encapsulation. For example, Deng [9] used two wrapping materials and their composites to wrap super-absorbent resin. The initial water-absorbent rate of super absorbent resin which was wrapped descended 17 times within 12 h, and the time of relieved swelling water was four days. The second one is composite cross-linking technology, which refers to the introduction of stable cross-linking agent and unstable cross-linking agent into the gel system. Under high cross-linking degree, the resin particles absorb water slowly, and temperature changes or other stimulating factors trigger the unstable cross-linking agent to decompose. The crosslink density is reduced, and the stable crosslinking agent can maintain the three-dimensional network structure of the gel to achieve the purpose of slow water absorption and swelling. Yu [10,11] synthesized a degradable cross-linked polymeric microsphere (DCPM) using polyethylene glycol(200) diacrylate (PEG-200 diacrylate) as unstable crosslinkers. When the aging temperature is higher than 100 °C, PEG begins to degrade. Bai [12] synthesized a temperature-resistant and high-strength polymer network. Based on retro Diels–Alder reaction, furfurylamine-2,2-bis(4-glycidyloxyphenyl)-propane oligomer (FA-BGPP) network can decompose into fragments at high temperature and exude to gel surface to form a lubricating layer, thus endowing Diels–Alder crosslinked gels with self-lubricating ability.

To achieve retarding swelling of PPGs and consider the temperature gradient in the formation, the thermosensitive hydrogels are conceived. Concerning thermosensitive hydrogels, two main volume transition types are identified (Figure 1) [13]. The first is that hydrogels exhibit a pyrocondensation property resulted from the thermosensitive polymers that have a lower critical solution temperature (LCST) in aqueous solution. For example, poly(*N*-isopropyl acrylamide) (PNIPAM) [14], poly(*N*,*N*-diethyl acrylamide) (PDEA) [15], poly(2-(*N*-(dimethylamino) ethylmethacrylate) (PDMAEMA) [16], poly(*N*-vinyl caprolactam) [17,18], etc., are LCST polymers. The second type of thermosensitive hydrogels refers to those with thermal expansion behavior resulted from the thermosensitive polymers that exhibit an upper critical solution temperature (UCST) in their phase diagram. The hydrogel is in collapse state at low temperature and swells when temperature increases above UCST. The UCST performance of the polymers can rely on Coulomb interactions (C-UCST) or hydrogen bonding (HB-UCST) [19,20]. C-UCST polymers are some zwitterionic polymers such as (poly-3-dimethyl(methacryloyloxyethyl) ammonium propane sulfonate (PDMAPS) [21,22] and poly(3-[*N*-(3-methacrylamidopropyl)-*N*,*N*-dimethyl] ammoniopropane sulfonate (PSPP) [23]) with UCST in water within the range of 0~100 °C. Commercially relevant HB-UCST polymers like poly (ethylene oxide) (PEO) [24,25,26,27,28], poly(vinylmethylether) (PVME) [29,30], modified pol(vinyl alcohol)s (PVA) [31,32,33], and poly(hydroxyethyl methacrylate) (PHEMA) [34] show both LCST and UCST behavior with an UCST higher than 100 °C. Polyacrylic acid and copolymers exhibit UCST behavior only at very high ionic strength (>400 mM of NaCl) or low pH (pH < 4) [35,36,37,38].

Compared with the C-UCST polymers, of which the UCST in water lower than 100 °C and the interchain associations of the zwitterionic groups were easily destroyed by the salt ions in saline [39], HB-UCST polymers can manipulate the volume transition higher than 100 °C relying on hydrogen bonding and are less sensitive to the salt ions. Among the HB-UCST polymers, hydrophobically modified poly(vinyl alcohol)s (PVA) can possess a UCST in the range of 100~200 °C depending on the modification. One example is partially butyl-functionalized PVAs [33] that samples with butyl contents of 7.6 or 9.6 mol% and a molar mass of 134 kg mol^−1^ showed a UCST of 103 °C or 108 °C, respectively. In order to prepare the retarding swelling PPGs with the volume phase transition temperature higher than 120 °C, a novel thermosensitive particle gels were developed using hydrophobically modified PVA in this work. The thermal expansion behavior of the as-prepared retarding swelling PPGs leads to the facile injection of the PPGs into high permeability zones and transportation into deep area through pore throats.

## 2. Experimental Section

### 2.1. Materials

Poly(vinyl alcohol) (PVA) samples with degree of polymerization of 1700 ± 50 and degree of hydrolysis (DH) values 99 mol% were obtained from Shanghai Titan Scientific Co., Ltd. (Shanghai, China) and used as received. Glutaraldehyde (GA, 50 wt.% aq. solution), hydrochloric acid (HCl, 36–38% wt.% aq. solution), and dimethyl sulfoxide (DMSO) were purchased from Kelong Chemical Co., Ltd., Chengdu, China, and used without further purification. The hydrophobic modifier (HPM) was developed by us. Distilled water was used as the solvent.

### 2.2. Synthesis of the Copolymer and Hydrogels

The procedure applied in this work was based on the method elaborated in previous studies [32,40]. The schematic synthesis procedure of the hydrophobically modified PVA hydrogel is shown in Figure 1. In a typical procedure, 90 mL of distilled water was placed into a 250 mL three necked round-bottom flask equipped with a cooler and dropping funnel. After adding 10.0 g of PVA, the reactor was placed into a water bath preheated to 95 °C and the stirring was maintained at 400 rpm for 2 h. 2 mL of the catalyst HCl (18 wt.%) was then added and the solution was thermostated at 15 °C. A predetermined amount of hydrophobic modifier (HPM) was added dropwise under vigorous stirring within 30 min. After 12 h the hydrophobically modified PVA copolymer were purified by dialysis bag (MWCO: 50000D, Spectrum™, Atlanta, GA, USA), which were denoted as HPVA.

To prepare retarding swelling hydrogels, a predetermined amount of glutaraldehyde as the cross-linker was added dropwise into the HPVA aqueous solution with stirring, and then the mixture was constant at 4 °C for 8 h. Afterward, the product was soaked in distilled water to eliminate the unreacted monomers and other impurities if any. The as-prepared hydrogel samples, labeled as CHPVA, were stored in an oven set at 60 °C for 24 h. The as-prepared CHPVA gels were named as CHPVA-*m*-*n*, in which *m* refers to [HPM] /[PVA] × 100 and *n* refers to [GA]/[PVA] × 100. The composition of a series of HPVA and CHPVA was listed in Table 1.

### 2.3. FTIR and ^1^H-NMR Measurements

Fourier transform infrared spectroscopy (FTIR) spectroscopic spectra were obtained on a Nicolet 6700 FTIR spectrometer (Thermo Scientific, Waltham, MA, USA) to monitor the changes that occurred in the characteristic groups of CHPVA hydrogels. The spectra were recorded at 4000–400 cm^−1^.

The ^1^H-NMR spectra were obtained with a 400 MHz Bruker Avance NEO600 spectrometer (Bruker, Billerica, MA, USA).

### 2.4. Thermogravimetric (TG) Analysis

Thermogravimetric measurements were carried out using a DSC823 TGA/SDTA 851e analyzer (Mettler Toledo Corp., Greifensee, Switzerland). Prior to the analysis, all the samples were completely dried at 60 °C for two days. In a typical test, 6.0 mg of sample was heated from 0 to 600 °C under nitrogen atmosphere at a constant heating rate of 10 °C/min. The weight loss of the sample as a function of imposed temperature was recorded by a data acquisition system equipped in the analyzer.

### 2.5. Scanning Electron Microscope (SEM) Imaging

The morphology of CHPVA samples was studied using a Quanta 450 scanning electron microscope (FEI, Hillsboro, OR, USA) at an excitation of 20 kV. A small amount of CHPVA samples was placed on a cooper grid and then evaporated using liquid nitrogen prior to observation. Afterward, SEM images with different magnifications were recorded.

### 2.6. Thermosensitive-Transparency

Changes in transmittance as a function of temperature were measured using a Light Transmittance Meter (LS116, Shenzhen Linshang Technology Co., Ltd., Shenzhen, China) equipped with a thermostatic bath (CD-200F, Shanghai Yiheng Scientific Instrument Co., Ltd., Shanghai, China) for aqueous polymer solutions (HPVA) and hydrogels (CHPVA) in a pressure vessel (filling volume: 35 mL, size: 26 × 125 mm). Ultrapure water was used as the reference. The LCST and UCST were defined as the temperatures where the transmittance dropped and increased to half of the original value, respectively. Changes in transmittance were recorded during the heating process, performed at a heating rate of 10 °C/h. The transmittance of HPVA solution and CHPVA hydrogels prepared with salinity water (21 × 10^4^ mg/L, Table 2) was measured.

### 2.7. Swelling Properties

The equilibrium swelling ratio (SR) of each sample was measured at room temperature (20 °C) by weighing fully swollen samples equilibrated in water. The hydrogels were dried at 60 °C for 12 h and then were ground into fine particles with size of 60–80 mesh. The gel particles were soaked in saline water (21 × 10^4^ mg/L) and SR was determined by the following equation (Equation (1)).
(1)$$SR=(m_{f}-m_{0})/m_{0}$$
where *m*_f_ is the final mass of the swollen gel at equilibrium swelling and *m*_0_ is the initial weight of the dry sample. The equilibrium SR was obtained from three parallel tests.

## 3. Results and Discussion

### 3.1. FTIR and ^1^H-NMR Measurements

The FTIR spectra of the GA, PVA, and CHPVA are shown in Figure 2. For the GA sample, characteristic peaks at 3420 cm^−1^ and 1727 cm^−1^ corresponded to the —C=O stretching. The peak at 2727 cm^−1^ and 1457 cm^−1^ corresponded to the —CHO bending and C—H stretching vibration characteristic absorption, respectively. The FTIR spectra show the typical peaks of —OH stretching vibration bands at 3442 cm^−1^ attributed to PVA and CHPVA. In contrast with the spectrum of PVA, the spectrum of the CHPVA gel presents a new absorption band at 1100 cm^−1^ (C—O—C group), demonstrating that the target gels were successfully prepared.

The structure of HPVA and the extent of the hydrophobic modification of PVA were determined by ^1^H-NMR spectra. The ^1^H-NMR spectra of HPVA and PVA are shown in Figure 3. The signal at 3.8 ppm and 1.5–1.2 ppm corresponded to the methine protons and the methylene protons in the main chains of both PVA and HPVA, respectively. The peak at 0.86 ppm in the spectrum of HPVA is attributed to the methyl group of the hydrophobically modified ring. The actual degree of hydrophobic modification of PVA can be estimated by calculating the peak area of the methyl proton and methylene proton, and HPVA-4 and HPVA-5 represent HPVA with 4 mol% and 5 mol% of hydrophobic modification, respectively.

### 3.2. Thermogravimetric Analysis

Figure 4 shows the TGA and DTG curves of PVA and CHPVA gels. The small mass loss of CHPVA gels in the temperature range of 120–200 °C can be attributed to the release of residual solvent molecules in the gel. The -OH group can be degraded in the temperature range of 200–300 °C, and the elimination of side groups is carried out when the temperature is higher than 300 °C. The residual amount of CHPVA gel at 500 °C is 15%, while the residual amount of pure PVA at the same temperature is only 7%, and it is almost completely degraded at 600 °C. It can be concluded that the thermal stability of CHPVA gel is enhanced compared to PVA. It can be seen from the DTG curve that the gel has a decomposition peak at 188 °C and 196 °C, respectively. Compared with the TGA curve, it is found that the mass of the CHPVA gel is reduced by about 10% at this time, which is related to the volatile impurities in the sample during heating. The results indicate that the CHPVA gels possess excellent thermal stability and can meet the requirements of underground high-temperature oil reservoirs.

### 3.3. Micromorphology of CHPVA Hydrogels

In order to maintain the swollen morphology of the hydrogels, fully swollen CHPVA hydrogels were treated by lyophilization. The SEM images of two gel samples (CHPVA-5-0.5 and CHPVA-4-0.5) are presented in Figure 5 to demonstrate the hydrogel network structure. As recognized, the mechanical and swelling properties of gels are highly dependent on their network structure. Swollen gels showing multiporous network structure is responsible for their water absorptivity. With the same chemical cross-linking density and different degrees of hydrophobic modification, the two CHPVA gels, i.e., CHPVA-5-0.5 and CHPVA-4-0.5, show no significant distinction between their micromorphology. The SEM characterization can corroborate the water absorptivity of the CHPVA gels.

### 3.4. Thermosensitive-Transparency

The temperature-induced phase transitions of aqueous solutions of HPVA with varied degrees of modification are shown in Figure 6a. The variations in LCST and UCST of HPVA with the degree of modification are summarized in Figure 6b. In the range of the hydrophobic modification of 4.0~6.0 mol%, the LCST of HPVA solutions decreases steeply from 71 °C to 51 °C with the increase in the degree of modification, whereas the LCST and UCST are not observed at very low hydrophobic modification (<4.0 mol%). The results are presumably due to the hydrogen bonding. With a low degree of hydrophobic modification, most of the hydroxyl groups of PVA are retained and the hydrophobic association between the modified PVA chains is very weak. Accordingly, the temperature increase is not able to lead to the phase separation of the HPVA solutions. Comparatively, with the increase in the hydrophobic modification of PVA, i.e., from 4.0 mol% to 6.0 mol%, the thermal induced hydrophobic association is enhanced and consequently the LCST of the HPVA solutions decreases. It is worth noting that the UCST of the HPVA with the modification degree of 4.0, 4.5, and 5.0 mol% was detected at 97 °C, 107 °C, and 112 °C, respectively. It is indicated that the temperature is high enough to provide the thermal energy sufficient to break the hydrophobic effect and hydrogen bonding between the modified PVA chains and to form solution polymer chains again. However, for HPVA with the modification degree of 5.5 mol% and 6.0 mol%, the hydrophobic association is too strong to be broken even when the temperature is higher than 130 °C.

The effect of crosslinking density of the hydrogels on the LCST and UCST behaviors was also investigated. In this vein, six CHPVA hydrogel samples with two different degrees of hydrophobic modification, i.e., 4.0 mol% and 5.0 mol%, were synthesized by varying the concentration of the crosslinker glutaraldehyde (0.5%, 1.0% and 1.5 wt.%). Figure 7 shows the plots of transmittance as a function of temperature for the hydrogels. It can be seen that for both CHPVA hydrogels with modification degree of 4.0 mol% and 5.0 mol%, the crosslinking density has a great effect on the thermosensitivity of the hydrogels. For the CHPVA hydrogels with the same modification degree, both the LCST and UCST increase with the increase in crosslinking density, indicating the increase in crosslinking density of CHPVA hydrogels require temperature increase to provide higher thermal energy to break the inter-/intra-molecular and polymer-water effect. In addition, the LCST of the chemically cross-linked hydrogels decreases whereas the UCST increases compared with the HPVA solution with the same degree of hydrophobic modification.

### 3.5. Swelling Properties

Figure 8 shows the swelling ratio of the two CHPVA hydrogels listed in Table 1 with the HPM/PVA molar ratio of 4.0% and 5.0% and the CPVA hydrogel with no hydrophobic modification as a function of temperature. Regarding the change in swelling ratio of the hydrogels, CPVA-0.5 does not show any volume phase transition temperature in the range of the test temperature. The slight decrease in SR of the hydrogel is due to the fact that the increase in temperature can break a certain amount of the hydrogen bonds inside the hydrogel and lead to water loss. With the increase in the degree of hydrophobic modification, the swelling ratio of the resulting hydrogel decreases and its volume phase transition temperature increases. The volume phase transition temperature of CHPVA-4-0.5 and CHPVA-5-0.5 hydrogels is 120 °C and 140 °C, respectively. The result implies that the higher the degree of hydrophobic modification, the less hydrogen bonds of PVA forms, resulting in the increase in UCST.

At a temperature lower than 60 °C, which approximates to the LCST of the hydrophobic-modified PVA hydrogels, both CHPVA-4-0.5 and CHPVA-5-0.5 hydrogels have relatively high swelling ratio. Comparatively, in the range of 60~120 °C which is between the LCST and UCST of the CHPVA hydrogels, CHPVA10-4-0.5 and CHPVA10-5-0.5 hydrogels show very low swelling ratio due to the soluble-insoluble phase transition. With the further increase in temperature, the swelling ratio of CHPVA hydrogels correspondingly increases resulting from the UCST phase transition. The soluble-insoluble-soluble phase transition property of HPVA polymer and swelling-shrinking-swelling change of CHPVA hydrogels generate the retarding swelling property of PPGs taking advantage of the temperature gradient. At low temperature (<60 °C), the injection speed of PPGs is high in the initial injection stage that the PPGs with high SR do not have enough time to swell to a great extent; at relatively high temperature (60~120 °C), the PPGs with low SR shrink and are able to migrate deeply; at high temperature (>120 °C for CHPVA-4-0.5 and >140 °C for CHPVA-5-0.5), the PPGs with increased SR reswell and achieve efficient water plugging.

On the other hand, the hydrogel with a lower degree of hydrophobic modification shows a higher initial swelling ratio. This observation may be explained by the intrinsic hydrophilicity of hydrogel based on PVA. The hydroxyl groups of PVA tend to form strong hydrogen bonding with water molecules. Therefore, with low hydrophobic modification degree and high PVA content, the modified hydrogels are hydrophilic even at low temperatures due to the strong hydrogen bonding with water molecules.

## 4. Conclusions

To address the high swelling speed and limited injection depth of conventional PPGs, we developed a novel temperature-induced retarding swelling and thermosensitive particle gel denoted as CHPVA. In this work, the preparation and thermosensitivity of CHPVA focusing on its swelling behaviors and thermal stability were investigated. Based on the experimental results, the following conclusions can be drawn:(1)A series of CHPVA hydrogels was successfully prepared, of which the structure was confirmed by FTIR and ^1^H-NMR. The micromorphology of the hydrogel network was presented by SEM imaging.(2)The measurement of light transmittance with temperature increase shows that the hydrophobic modification of HPVA will reduce the LCST of the polymer solution and increase the UCST. The CHPVA hydrogels with the same crosslinking density present the same tendency. For the CHPVA hydrogels with the same modification degree, both the LCST and UCST increase with the increase in crosslinking density.(3)The temperature-increasing swelling experiment shows that the swelling-shrinking-swelling change of CHPVA hydrogels with temperature generate the retarding swelling property of PPGs taking advantage of the temperature gradient and is responsible for the deep migration and efficient water plugging of PPGs.


## Data Availability

Not applicable.

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
