# Peer review of "LCST-UCST Transition Property of a Novel Retarding Swelling and Thermosensitive Particle Gel"

_materials, 2023, doi:10.3390/ma16072761_

Round 1

Reviewer 1 Report

The authors present a study on hydrophobically modified PVA hydrogels crosslinked with glutaraldehyde. Dependent on the degree of modification achieved hydrogels show a combined UCST/LCST behavior.

I suggest publication after minor revisions.

Some remarks:

1. The paper is difficult to read. One of the reasons is the lack of proper introduction of abbreviations. It starts with "CHPVA" which is not defined in the abstract. Also later in the text it is never explained why "C" denotes the hydrogel (although one could guess it is derived from "crosslinked"). All abbreviations have to be introduced, this also includes references to the work of other groups (e.g. in lines 33, 34, 46, and 48) and to methods applied (e.g. TGA, DTG)

2. Sample names should be as easy as possible in order to follow the story. In this regard the "10" as part of the sample name in all samples discussed in the paper (table 1) might be omitted.

3. The total of the ions given in table 2 does not sum up. Also, why is it numbered?

4. In figure 2 and 3 y-axis is not denoted. In figure 2 x-axis should be denoted with "wavenumber". Around figure 2 there is some confusion between "GA" and "GD". Which CHPVA sample is shown in figure 2?

Author Response

有关答复,请参阅附件。

Reviewer 2 Report

The authors submitted “LCST-UCST Transition Property of a Novel Retarding Swelling  and Thermosensitive Particle Gel”.  To publish in “Materials”. The reviewer thinks that The novelty of this work is interesting to publish in this journal, but some important data are missing. Here are my comments:

1. The reviewer suggests that the reviewer should provide the TOC of this work.

2. The schematic scheme of the copolymers in this study should be provided.

3. how do the authors investigate the molecular weight of these materials and their purity of these materials? GPC data should be provided.

4. In FTIR figures, please assign each band inside the figure. The same thing in the 1H-NMR figure.

5. the 1H-NMR data should be completed with 13C-NMR data.

6. to prove the LCST-UCST Transition Property of these materials, TEM and DLS and images in solution should be provided.

7. scale bar should be clear in all SEM images.

Reviewer 3 Report

The study under review is devoted to the synthesis of new super sorbent materials, which can be used in oil industry. New materials are cross-linked gels based on  hydrophobically modified poly(vinyl alcohol)s.

The work is carried out quite accurately including description of the technical operations in the synthesis and using modern method of synthesis and chemical structure characterization. Measuring the temperature dependencies of swelling ratio shows that this parameter can reach 10 (at room temperature) this is a good result.

I do not have serious objections. The only problem is a lot of abbreviation that makes reading the article a rather complicated work. I believe that abbreviations in the title of the article are unacceptable and it is desirable to escape them along the text

Round 2

Reviewer 2 Report

The authors did not do the reviewer's suggestions such as:

1. In FTIR figures, please assign each band inside the figure. The same thing in the 1H-NMR figure.

  2. . to prove the LCST-UCST Transition Property of these materials, TEM and DLS and images in solution should be provided.

 3. scale bar should be clear in all SEM images.
